# Anti-Condensation Performance of a New Superhydrophobic Coating for Pavements

**DOI:** 10.3390/ma16175793

**Published:** 2023-08-24

**Authors:** Kaijian Huang, Ruiyu Sun, Jiaqing Wang, Xijun Shi, Hechang Lei

**Affiliations:** 1College of Civil Engineering, Nanjing Forestry University, Nanjing 210037, China; sunruiyuyu@163.com (R.S.); jiaqingw@njfu.edu.cn (J.W.); 2Ingram School of Engineering, Texas State University, San Marcos, TX 78666, USA; xijun.shi@txstate.edu; 3Beijing Key Laboratory of Opto-Electronic Functional Materials & Micro-Nano Devices, Department of Physics, Renmin University of China, Beijing 100872, China; hlei@ruc.edu.cn

**Keywords:** superhydrophobic coating, contact angle, pavement, anti-condensation

## Abstract

Superhydrophobic coating ice suppression is an advanced and durable technology that shows great potential for application on pavements. Although many researchers have conducted experimental and theoretical validations to confirm the effectiveness of superhydrophobic surfaces in actively suppressing ice formation, there are still some who remain skeptical. They argue that the roughness of the surface may increase ice adhesion due to the mechanical interlocking effect of condensation droplets in low-temperature and high-humidity environments. In this study, we present a comprehensive investigation of a novel superhydrophobic coating specifically designed for pavement surfaces, aiming to address the question of its active anti-icing/ice-sparing capabilities in a condensing environment. The changes in contact angle before and after condensation for four material surfaces with varying wettability were investigated, as well as the morphology and ice adhesion of liquid water after it freezes on the material surface. The findings reveal that the proposed superhydrophobic coating for pavements effectively prevents condensate droplets from infiltrating the surface structure, resulting in delaying the surface icing time and reducing the attachment strength of the ice.

## 1. Introduction

The accumulation of snow and ice on the pavement surface in winter leads to a significant reduction in the skid resistance of the pavement surface, seriously weakening road capacity and posing a grave threat to vehicle safety, potentially leading to severe traffic accidents [1,2]. This threat is particularly prominent in southern China, where freezing rain and black ice form in high-altitude, cold, and wet areas [3,4], further jeopardizing road safety. Therefore, addressing the issue of snow and ice on pavements during winter is crucial to ensure traffic safety.

Currently, two main types of anti-icing technologies are employed for pavements, namely active de-icing technology and passive de-icing technology. Passive de-icing technology primarily relies on physical or chemical methods such as manual mechanical removal and the application of snow-melting agents. Although this approach is relatively efficient, it still has the potential to damage the pavement surface and bring about pollution to the surrounding ecological environment. What is more, the costs associated with passive de-icing methods are higher [5,6]. In contrast, active de-icing methods are more efficient and less damaging to pavement structures. Examples include self-heating pavements, pavements containing anti-freeze fillers, phase change energy storage pavements, carbon fiber conductive pavements, and emerging superhydrophobic materials [7,8,9,10,11]. However, the currently available active de-icing methods in the market are often not affordable. Notably, the utilization of superhydrophobic materials, serving as a potential active de-icing solution, has garnered increasing attention due to its simple nature of preparation, cost-effectiveness, and, more importantly, free from environmental contamination [12,13].

There are several studies [14,15] that have investigated ice adhesion mechanisms on superhydrophobic surfaces. In 2003, Pilotek [16] first proposed that superhydrophobic coatings might exhibit low ice adhesion. It promotes the application of superhydrophobic surfaces in the field of anti-icing coverage. Some researchers [17,18] have compared ice adhesion for different surface structures and concluded that nanostructured surfaces with relatively lower roughness exhibit the lowest ice adhesion. Subramanyam’s group at MIT, USA [19] fabricated four different surface structures, including smooth, micron, nano, and micro/nanocomposite surfaces, modified with surface energy substances. Their analysis revealed that the nanostructured and micro/nanocomposite surfaces exhibited excellent resistance to frost formation, with the nanostructured samples displaying the lowest ice adhesion. This finding has provided significant inspiration. Huang’s non-uniform nucleation ice crystal growth model experimentally verifies the ice inhibition mechanism of superhydrophobic materials. The results show that the prepared superhydrophobic coating material exhibited excellent properties, including a high contact angle (>150°), good anti-slip nature, and the ability to retard water droplet crystallization and maintain the droplet shape even after freezing [20]. These findings highlight that superhydrophobic surfaces achieve extremely low surface energy due to the presence of micro-nano rough structures. Also, the grooves on the substrate surface are filled with air, resulting in a Cassie state for droplets, where only about 10% of the total contact area is occupied by the droplet and substrate contact [21]. This, combined with the surface’s exceptionally low rolling angle, hinders the infiltration of droplets into pavement material upon impact, effectively reducing the icy surface area of the pavement. This, in turn, reduces the area of ice in contact with the pavement. The adhesion between the ice and the pavement material is reduced.

The surface affects the wettability of superhydrophobic coatings in terms of ice adhesion remains controversial. Some researchers argue [22,23,24,25] that ice adhesion is linked to surface wettability and studied the relationship between surface wettability and ice adhesion force. Their findings indicate that ice adhesion force is proportional to smaller hysteresis angles, resulting in lower adhesion forces. While some researchers hold the opposite opinion [26,27,28,29,30], for instance, Chen [31] prepared 13 kinds of silicon wafers with superhydrophilic or superhydrophobic wettability to study surface morphology and adhesion. It was found that ice adhesion on superhydrophilic and superhydrophobic surfaces was similar. Chen’s research concluded that superhydrophobic coatings not only fail to reduce ice-cover bond strength but also exacerbate subsequent de-icing efforts due to their surface structure and mechanical interlocking effect between ice and the coating. Varanasi [32] investigated the application of superhydrophobic surfaces for de-icing and revealed that rough superhydrophobic surfaces in high humidity environments are susceptible to frost formation, therefore resulting in greater ice adhesion compared to smooth surfaces. These results raise questions about the effectiveness of superhydrophobic surfaces for de-icing applications.

Actually, superhydrophobic surfaces are frequently exposed to low-temperature and high-humidity environments, where gas–liquid phase change condensation or fog coalescence, commonly observed in nature, inevitably occurs on solid surfaces [32]. As the supercooling increases, water vapor condenses and nucleates within the micro- and nano-structures of the superhydrophobic surface and grows gradually. Failure to expel the condensed droplets from the surface irreversibly transforms the surface into a partially or completely wetted Wenzel state, adhering to the surface [33,34,35]. At this stage, when external droplets come into contact with the condensed surface, they fuse with the droplets present on the surface, causing a reduction in the apparent contact angle and altering the surface wettability. Consequently, the superhydrophobic ice-suppressing performance also diminishes. Therefore, to effectively utilize superhydrophobic ice-suppressing coatings on pavements, theoretical discussions and feasibility studies regarding the anti-condensation performance of superhydrophobic materials are valuable, ensuring the desired active anti-icing/ice-sparing effects.

This paper employs the “self-migration” movement of condensate droplets on superhydrophobic coatings to theoretically analyze factors influencing the anti-condensation performance of the coating. We characterize four material surfaces with varying wettability to investigate changes in the contact angle of different material surfaces before and after condensation and the morphology and ice adhesion of liquid water after freezing on the material surface to support the proposed theory. Condensed droplets do not affect the water contact angle of the superhydrophobic surface, and the superhydrophobic surface can delay the icing time of the droplets, as the droplets have a smaller contact area and adhesion force with the material surface after icing. The superhydrophobic coating has good anti-condensation properties and broad application prospects in active ice suppression on pavements.

## 2. Theoretical Analysis of Ice Suppression Performance of Superhydrophobic Surfaces due to Condensation

The findings from recent investigations [36,37,38,39,40,41] demonstrate that on certain nanostructured superhydrophobic surfaces, condensate droplets exhibit self-migration behavior or jumping behavior without gravitational influence. The mechanism behind the phenomenon is based on the release of surface energy into kinetic energy. Yet, the adhesion of the droplets to the superhydrophobic surface is a critical factor regulating the self-migration of droplets, specifically the adhesion between the droplets and the surface. Smaller interaction forces facilitate easier self-migration. The interaction force is minimized when the droplets are in the Cassie state, enabling self-migration. Even without self-migration, droplets in the Cassie state can easily roll off the surface under an external force, thus keeping the surface dry.

This “self-migration” of condensate droplets provides a new strategy for ice suppression: when the surface of a superhydrophobic material is in the Cassie state, the condensate droplets can jump off the surface without freezing in the microstructure of the material. The main processes include water vapor condensation nucleation, nucleation growth, merging and polymerization, self-jumping, and departing the surface. Therefore, when preparing a superhydrophobic ice-suppressing surface, as long as the Cassie state is maintained when condensate droplets are present on the material surface, these condensate droplets will not exist in the material microstructure, and with the occurrence of the “self-migration phenomenon”, the condensate droplets remain unaffected, preserving the ice-suppressing effect. 

Among these processes, water vapor condensation nucleation involves the aggregation of water vapor molecules and is typically induced by factors such as water vapor supersaturation and nucleation-inducing agents like dust and nanostructures (heterogeneous and homogeneous nucleation). The critical nuclear radius represents the minimum size at which water vapor molecules aggregate into stable droplets. It depends primarily on the supersaturation resulting from the dew point, subcooling temperature, and relative humidity. According to classical nuclear theory and previous research [42,43], the critical nuclear radius can be calculated using the following equation:(1)$$r_{c}=-\frac{2\gamma v}{\Delta G}$$

Based on the nucleation theory, the minimum radius at which a droplet can nucleate on a superhydrophobic surface can be calculated. The interfacial tension of water (0 °C) is $\gamma =7.56\times 10^{-2}\;J/m^{2}$, the molar volume of water molecules is $v\approx 1.8\times 10^{-5}\;m^{3}/\mathrm{mol}$, the Gibbs Free Energy is $\Delta G\approx -\mathrm{CpT}\left[\ln\left(T/\mathrm{Td}\right)+\mathrm{Td}/T-1\right]$, T = 273.15 K (0 °C), water dew point temperature vapor (70%RH, 25 °C) is Td = 291.15 K (15 °C), and the specific heat capacity of water vapor is C_p_ = 33.5 J/mol·K. Finally, the nucleation on the superhydrophobic surface can be obtained for about 145 nm.

The critical nuclear radius represents the smallest droplet size that forms a stable nucleus, and in most cases, water vapor undergoes heterogeneous nucleation upon contacting the surface. Consistent with the critical radius is the free energy barrier, which is another factor describing the ease of nucleation, especially for micro- and nano-structured surfaces. According to classical nucleation theory, the influence of surface structure on the heterogeneous free energy barrier of condensed water droplets can be analytically determined using the following equation:(2)$$\Delta G_{c}=\Delta G_{c}^{\mathrm{homo}}f\left(m,x\right)$$
where $\Delta G_{c}^{\mathrm{homo}}$ is the free energy barrier in the homogeneous nucleation of water droplets. $f\left(m,x\right)$ is the ratio of the nucleation-free energy barrier of a spherical water droplet relative to its volume. Since the critical radius of 145 nm is the definite value, $f\left(m,x\right)$ varies only with the radius ($R_{s}$) of the surface structure.

When water droplet nucleation occurs in nano-gaps that are less than 145 nm wide, the small nucleation barrier energy allows the nuclei to grow and merge into larger, micron-sized droplets. These droplets then undergo spontaneous upward movement to reach the top of the surface structure, forming the Cassie state. As time passes, the condensed droplets on the horizontally placed superhydrophobic surface begin to fuse and undergo self-migration, leading to the appearance of micron-sized water droplets on the surface. As such, water vapor tends to nucleate on the micron-sized large water droplets rather than in the gaps of the nanostructure because the gaps are nearly the same size or smaller than the critical nuclear radius of water droplets, bringing about high nuclear barrier energy. In this way, the condensate drops will not enter the nanometres of the superhydrophobic surface with a width less than the critical nuclear radius. This prevents the development of a mechanical interlocking effect, and the ice adhesion of the surface does not increase. Therefore, even in low temperature and high humidity environments, as long as the surface microform meets the requirements, the liquid water on the surface of the superhydrophobic coating will remain in a Cassie state and will not be affected by the condensation phenomenon. The superhydrophobic surface will maintain the droplet hemispherical shape after the liquid water freezes, and the ice adhesion will be smaller than that of the ordinary smooth surface.

## 3. Experiment

### 3.1. Sample Preparation

#### 3.1.1. Testing Raw Materials

In this paper, four surfaces with different wettability were prepared for various experimental tests using the following raw materials (Table 1).

#### 3.1.2. Preparation Steps

In this paper, hydrophilic surfaces were prepared using SiO_2_. Si(OH) sols were synthesized by mixing tetraethyl orthosilicate and hydrochloric acid solution at a mass ratio of 2.5:1. The pH was adjusted to 2.5, and the mixture was magnetically stirred for 2 h at room temperature. The specification of the magnetic mixer is DF-101S. The Si(OH) sol was then combined with a silica suspension, and the resulting mixture was stirred magnetically for 30 min. Subsequently, it was dispersed using an ultrasonic disperser for another 30 min to obtain the coating solution (Figure 1). 

Furthermore, based on the principles of hydrophobicity, surface modification of nanomaterials was conducted using low surface energy substances to prepare a hydrophobic surface with nanosilica. The silica particles and stearic acid were dispersed in anhydrous ethanol, sonicated, and magnetically stirred at room temperature. The heating reaction of stearic acid with anhydrous ethanol produces ethyl stearate with good lubricity and thermal stability, and the low surface energy substance produced by the reaction of stearic acid with ethanol constructs the rough structure of superhydrophobic materials. The hydrophobic modification effect of stearic acid on titanium dioxide nanoparticles played a decisive role in the hydrophobicity of the samples. The mixed solution was centrifuged to stabilize it and then dried to obtain hydrophobic nanomaterials (Figure 2). The specification of the centrifuge is TGL-16A, provided by Hunan Xiangli Equipment Co., Ltd. (Changsha, China).

The specific steps for the preparation of superhydrophobic materials are as follows: to prepare the TiO_2_-SiO_2_ composites, titanium dioxide nanopowder was initially dispersed into anhydrous ethanol through sonication. Subsequently, silicon dioxide and stearic acid were added to the initial solution. The mixture was then subjected to sonication and stirring at 70 °C for 6 to 12 h using a magnetic stirrer. Following this, the mixture was centrifugally dried and ground to obtain a superhydrophobic material powder; the process is shown in Figure 3. Finally, the hydrophilic, hydrophobic, and superhydrophobic materials were sprayed onto the surface of the material using a PQ-2 spray gun, provided of Shanghai Xuling Information Technology Co., Ltd. (Shanghai, China)

### 3.2. Test Methods

#### 3.2.1. Apparent Morphology

The morphology and particle distribution of the samples were observed using a Regulus 8100 field emission scanning electron display microscope, provided by Shanghai Na Scientific Instrument Co., Ltd. (Shanghai, China). The sample coated with superhydrophobic material was firmly secured to the sample stage with a conductive adhesive, ensuring that the viewing surface was facing upwards. Then, after the conductive adhesive was air-dried, a thin layer of gold was sprayed on the surface of the sample to ensure good electrical conductivity [44]. The magnification of the sample morphology could be adjusted accordingly.

#### 3.2.2. Water Contact Angle Test

Four different surfaces with varying wettability were designed for this paper: hydrophilic surface, hydrophobic surface, superhydrophobic surface 1 (made from modified anatase titanium dioxide), and superhydrophobic surface 2 (made from modified rutile titanium dioxide). The contact angles produced by condensate (at 0 °C, humidity RH = 70%) were measured in a room temperature environment (25 °C, humidity RH = 30%) both before and after condensation (with a temperature difference of over ten degrees Celsius). As the temperature decreased and humidity increased, the contact angles of the characterized surfaces changed. They were carefully deposited onto the upper sample surface using a syringe. The droplet mass was 5 mL. A contact angle tester (Data Physics OCA40Micro, provided of Beijing Oriental Defei Instruments Co., Ltd., Beijing, China) was used to measure the static contact angle of the drop on the sample surface. Five different positions on the surface of the material were selected for contact angle measurement, and their average values were calculated.

#### 3.2.3. Freezing Experiments

To simulate the icing morphology of water droplets under low temperature and high humidity conditions, drip icing tests were conducted on superhydrophobic coatings at temperatures of −4 °C, −8 °C, and −10 °C with a relative humidity of 70%. The specific implementation steps were as follows: Three specimens with identical dimensions were prepared for ordinary uncoated specimens, hydrophobic coated specimens, and superhydrophobic coated specimens 1 and 2, respectively.The sample was placed in a temperature control box, and 5 mL of water droplets were carefully deposited onto the surface of each specimen. The morphology of the frozen water droplets was observed. To minimize operational errors, five sets of experiments were conducted for each condition.

#### 3.2.4. Ice Adhesion Strength Test

The adhesion of ice on pavement surfaces plays a critical role in road traffic safety, as the presence of snow and ice can significantly increase the bond size between the ice and the pavement, making de-icing efforts challenging. Since spray coating on the pavement surface can alter the original characteristics of the pavement, it is essential to evaluate the ice adhesion of novel superhydrophobic coatings after icing. The experimental schematic diagram is presented below, along with specific test steps (Figure 4).

Prepare concrete specimens measuring 100 mm × 100 mm × 100 mm in advance and place them into a mold measuring 400 mm × 100 mm × 100 mm. Separate the two specimens with a plastic film, leaving a 100 mm gap at each end.Fill the empty slots at both ends with water, ensuring that the water comes into contact with the side of the specimen that will be used for the test surface. Transfer the mold to a freezer set at a temperature of 10 °C. After 4 h, remove the mold from the freezer. At this point, the specimens and water will have frozen into two rectangular iced concrete specimens measuring 200 mm × 100 mm × 100 mm.Demold the frozen concrete specimens and place them onto a press. Position two arc-shaped pads at the interface between the specimens and the ice on both ends. Prior to use, ensure that the arc-shaped pads are subjected to the same freezing conditions as the specimens, ensuring that pads with different temperatures do not impact the bond between the ice and the concrete specimens. Apply pressure to the specimen using the HTC-1068 pressure tester (provided by Beijing Fuhaida Technology Co., Ltd., Beijing, China) until splitting occurs.Record the results and compile the data.

## 4. Results and Discussion

### 4.1. Microscopy of Superhydrophobic Materials 

The micromorphology of the superhydrophobic material was examined using FESEM, and the results are presented in Figure 5. During the hydrothermal reaction, SiO_2_ particles were observed to encapsulate the surface of TiO_2_ particles. The nano-TiO_2_ reacted with stearic acid, leading to a rough morphology of the nanostructures and the formation of a large number of papillae. The lotus leaf is a natural superhydrophobic surface due to its special micro-nanostructure and self-cleaning characteristics. It can be observed in the electron micrograph that the superhydrophobic surface has the same micronano papillary structure as the lotus leaf surface. Thus, the field emission scanning electron microscopy (FESEM, provided of Shanghai Sinu Optical Technology Co., Ltd., Shanghai, China) analysis confirmed the presence of superhydrophobic materials with low surface energy.

### 4.2. Effect of Condensation on the Water Contact Angle of Materials

The contact angle test was performed to evaluate the wettability of water droplets on the surface of the superhydrophobic material. Multiple measurements were taken at different locations on the coated sample. A surface material can be classified as superhydrophobic when the contact angle of water droplets exceeds 150°. The contact angle results are summarized in Table 2.

The results show that the water drop contact angle on the prepared superhydrophobic coating 2 was 154.0° (>150°) at room temperature and 151.9° (>150°) after the condensation phenomenon. These findings demonstrate that the coating material still has a large water contact angle at low temperatures and high humidity, conforming to the characteristics of wettability as superhydrophobicity, thereby confirming its robustness in maintaining superhydrophobicity.

### 4.3. Water Droplet Icing on Different Wettability Surfaces

Following the freezing test procedure described in Section 3.2.3, the samples were placed in a temperature control box set to −4 °C. After 20 min, the droplets on the surface of the normal sample were observed to freeze, while those on the hydrophobic coating exhibited a mixed state of ice and water. Remarkably, the droplets on the superhydrophobic coating remained in a liquid state. As time progressed, after 35 min, the droplets on the superhydrophobic coating eventually froze. This indicates that at a temperature of −4 °C, the solidification time of droplets on superhydrophobic surfaces was significantly longer compared to the ordinary specimen surface, thus demonstrating the ability of the superhydrophobic coating to extend the solidification time of droplets. Next, the temperature of the high-temperature control box was changed to −8 °C, and another set of comparison specimens was placed inside. It was observed that the droplets on the surface of the normal specimen started to freeze after about 8 min, while the water droplets on the hydrophobic coating surface began freezing after about 11 min. In contrast, the water droplets on the superhydrophobic coating surface exhibited freezing after approximately 14 min. Finally, a final group of specimens was placed in the high and low-temperature control box set to −10 °C to observe the time required for water droplets to freeze. The results indicated that the droplets on the regular specimen surface started freezing after about 3 min, the droplets on the hydrophobic coating surface started freezing after approximately 6 min, and the droplets on the superhydrophobic coating surface started icing after approximately 8 min. Figure 6 shows the photos of water droplet states on the regular specimen surface and the superhydrophobic-coated specimen at the same moment. Among them, Figure 6A–C represent the photos of droplets freezing on the surface of the regular specimen, Figure 6D–F represent the photos of drops of water on the surface of the hydrophobic coating specimen, and Figure 6G–I represent the photos of droplets on the surface of the superhydrophobic coating specimen.

Analyzing the figures, under −4 °C conditions, Figure 6A shows the state when the droplets on the regular specimen surface start to freeze, with small scattered ice crystals already present. Figure 6D represents the hydrophobic coating specimen at the same moment, while Figure 6G depicts the superhydrophobic coating specimen, where the surface water droplets remained in a liquid state. Under −8 °C and −10 °C conditions, Figure 6B,E,H represent the state after freezing at the same moment, while Figure 6C,F,I represent the state after icing at the same moment. It can be observed that the icing time of Figure 6B,C is shorter, while the icing time of Figure 6E,F,H,I is longer. Additionally, the ice droplet contact area with the wall of the dish after icing was larger in both regular specimens and hydrophobic coating specimens compared to the superhydrophobic coating. The water droplets in Figure 6G–I gradually increased in size as the temperature decreased, suggesting nucleation of water vapor condensing on the large droplets in a low-temperature and high-humidity environment.

### 4.4. The Law of Ice Adhesion Cover on Different Surfaces

To evaluate the ice cover adhesion of the new pavement superhydrophobic coating after icing, specimens of hydrophilic coating, hydrophobic coating, superhydrophobic coating modified with anatase titanium dioxide nanoparticles, and superhydrophobic coating modified with rutile titanium dioxide nanoparticles were prepared, corresponding to No. 1, No. 2, No. 3, and No. 4, as shown in Figure 7a. These specimens were placed in molds, as depicted in Figure 7b, inside high- and low-temperature control boxes set to −10 °C, allowing the water to condense into rectangular iced concrete specimens of dimensions 200 mm × 100 mm × 100 mm.

The iced concrete specimens were positioned on a press, and two circular mat strips were inserted at both ends of the interface between the specimens and the ice. Pressure was applied until splitting occurred, and the experimental results are presented in Figure 8, illustrating the ice adhesion to the four specimens. The flat hydrophilic surface exhibited the highest ice adhesion, measuring 947.75 N. This is due to the smoothness of the hydrophilic surface, which results in a larger actual contact area between the ice surface and the specimen. The ice adhesion of the hydrophobic surface and the superhydrophobic surface modified with rutile titanium dioxide was below that of the smooth hydrophilic surface, measuring 252.75 N and 237.75 N, respectively. Remarkably, the ice adhesion of superhydrophobic surfaces modified with rutile titanium dioxide was as low as 214 N. This can be attributed to the enhanced stability and superior weather resistance of rutile titanium dioxide compared to anatase titanium dioxide nanoparticles. Consequently, the impact of environmental changes on the superhydrophobic surface was minimized. 

For superhydrophobic surfaces, at room temperature, liquid water resided above the surface texture, known as the Cassie state, while the air trapped in the surface structure beneath the liquid water remained in thermodynamic equilibrium with it. As the temperature decreased, the surface liquid water gradually began to freeze, and the condensed droplets of air did not freeze within the superhydrophobic surface texture where the nano-gap fell below the critical nuclear radius. Instead, they condensed and nucleated on the surface of larger droplets. At this stage, the liquid water existed in a state of coexistence between Wenzel and Cassie states, as shown in Figure 9. As a result, the actual contact area between the ice surface and the sample decreased, resulting in a reduction in the ice adhesion.

## 5. Conclusions

Applications of superhydrophobic coatings as anti-icing materials for pavements can effectively delay the icing of static water before icing and reduce the adhesion of ice to the pavement surface after icing due to their unique micro- and nano-structures and excellent anti-condensation properties. Through both theory and experimentation, the study yielded important conclusions about the mechanisms behind these coatings:(1)Through theoretical discussions, we have uncovered the ice suppression capabilities of the superhydrophobic surface in relation to the condensate. It can be concluded that when the nano-gap of the superhydrophobic surface texture is lower than the critical nuclear radius of 145 nm under low humidity and high moisture environment, it prevents condensate droplets from entering the surface texture, allowing liquid water to remain in the Cassie state on the superhydrophobic surface. This inhibits the formation of mechanical interlocking effects, which can increase ice adhesion on the surface.(2)The surface of the superhydrophobic coating is composed of micro- and nano-papillae. Even in conditions of low temperatures and high humidity, condensation on the superhydrophobic surface does not affect its water contact angle, which is 151.9° with little change compared to the pre-condensation angle of 154.0°, ensuring continued strong superhydrophobicity. The large contact angle conceals the air in the gap of the coarse structure, which delays the icing time of the droplets and reduces the contact surface of the droplet with the superhydrophobic coating.(3)After investigating the adhesion of ice on a variety of wettability surfaces, our findings revealed that the adhesion force of hydrophilic surfaces with ice is the most robust, up to 947.75 N. In contrast, it can be observed that the presence of air within the microstructure of superhydrophobic surfaces significantly reduces the ice adhesion, with an adhesion of 214 N, without any observable mechanical interlocking effect.

## Figures and Tables

**Figure 1 materials-16-05793-f001:**
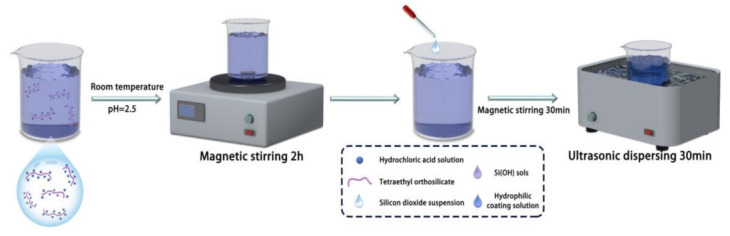
Preparation of hydrophilic coating.

**Figure 2 materials-16-05793-f002:**
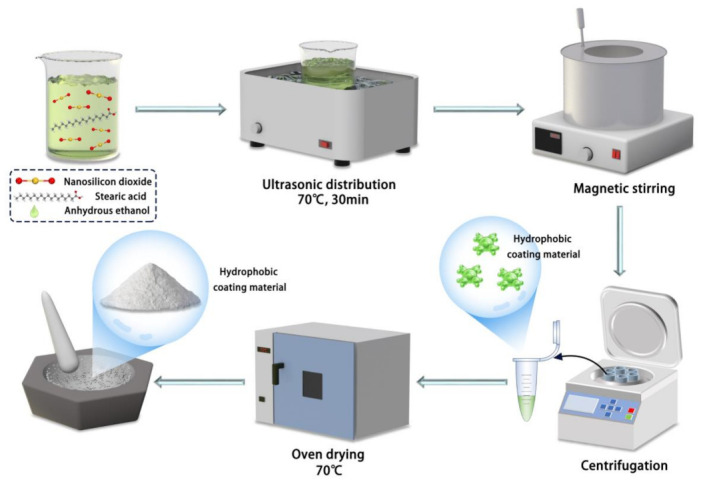
Preparation of hydrophobic coating.

**Figure 3 materials-16-05793-f003:**
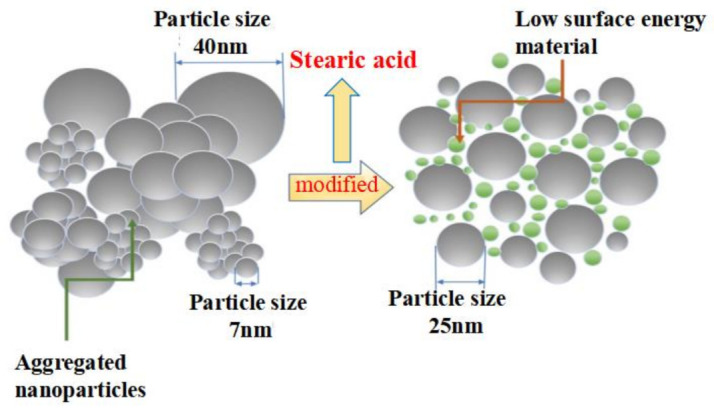
Preparation of superhydrophobic coatings.

**Figure 4 materials-16-05793-f004:**
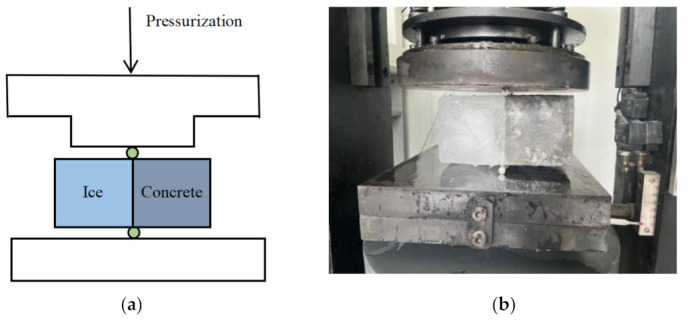
Ice adhesion strength test. (**a**) Schematic diagram of test principle. (**b**) Iced concrete specimen with pressure splitting diagram.

**Figure 5 materials-16-05793-f005:**
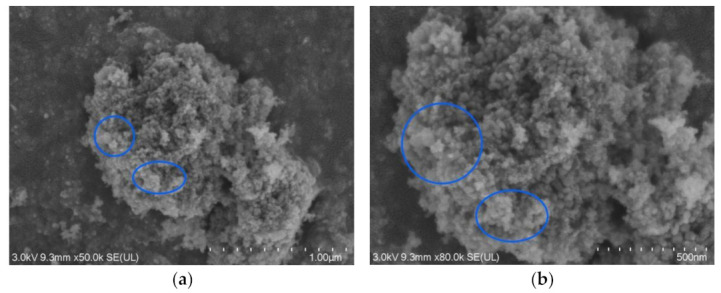
Field emission scanning electron microscopy images of the superhydrophobic surface modified with rutile titanium dioxide nanoparticles. (**a**,**b**) Electron microscopy images at different magnifications: images (**a**,**b**) are under ×50,000 and ×80,000 magnification scales, respectively.

**Figure 6 materials-16-05793-f006:**
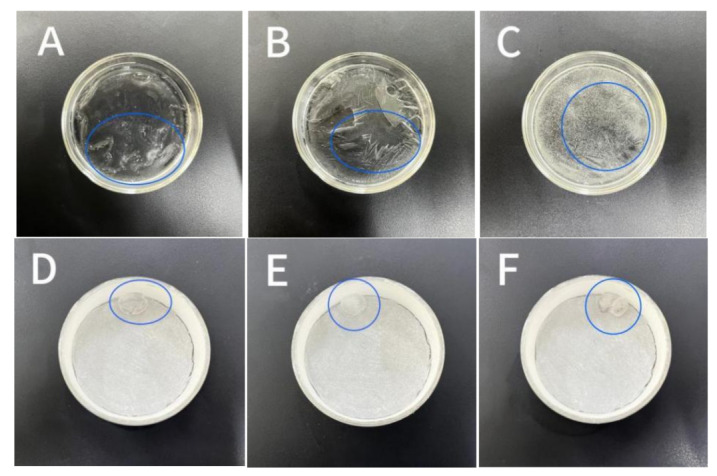
After the water droplets freeze on the sample surface. (**A**–**C**) Photos of droplets freezing on the surface of the regular specimen; (**D**–**F**) photos of drops of water on the surface of the hydrophobic coating specimen; (**G**–**I**) photos of droplets on the surface of the superhydrophobic coating specimen.

**Figure 7 materials-16-05793-f007:**
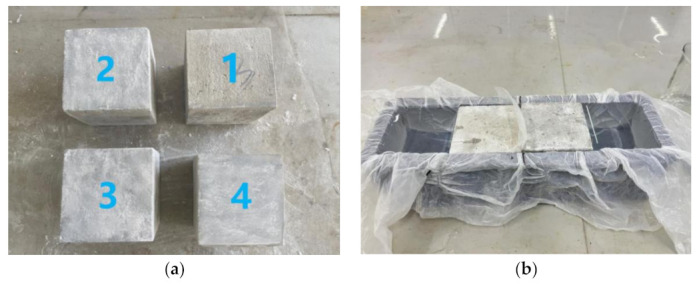
Sample preparation: (**a**) experimentally prepared hydrophilic coating specimen, hydrophobic coating specimen, superhydrophobic coating specimen; (**b**) ice adhesion strength test sample making mold diagram.

**Figure 8 materials-16-05793-f008:**
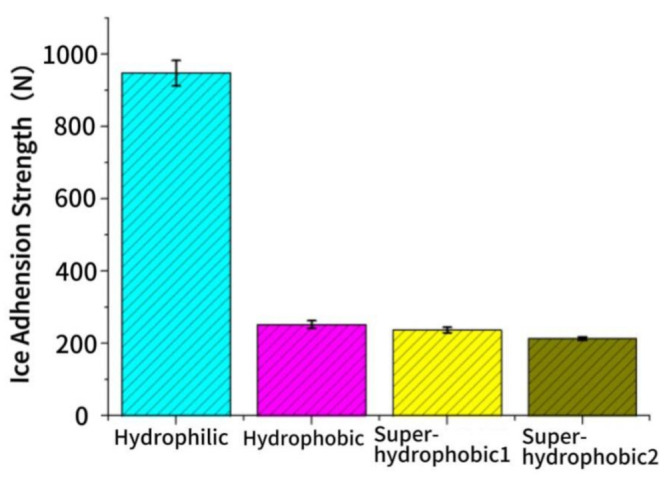
Adhesion strength of ice cover.

**Figure 9 materials-16-05793-f009:**
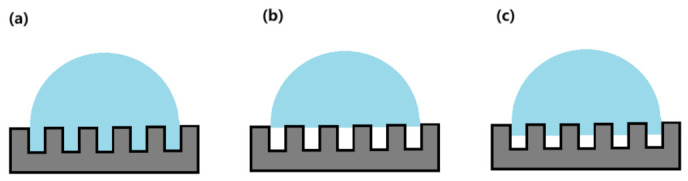
Mechanism of ice suppression on superhydrophobic surfaces: (**a**) Wenzel model; (**b**) Cassie model; (**c**) Wenzel and Cassie coexistence model [45].

**Table 1 materials-16-05793-t001:** Main test reagent specifications and manufacturers.

Reagent Name	Specification	Manufacturers
tetraethyl orthosilicate	98%	Sinopharm Chemical Reagent Co., Ltd. (Shanghai, China)
hydrochloric acid solution	0.01 mol/L	Shanghai Da Biotechnology Co., Ltd. (Shanghai, China)
silicon dioxide	7–40 nm	Sinopharm Chemical Reagent Co., Ltd.
stearic acid	AR	Sinopharm Chemical Reagent Co., Ltd.
anatase-structured titanium dioxide	25 nm	Sinopharm Chemical Reagent Co., Ltd.
rutile-structured titanium dioxide	60 nm	Sinopharm Chemical Reagent Co., Ltd.
anhydrous ethanol	AR	Sinopharm Chemical Reagent Co., Ltd.
deionized water	hyperpure	Laboratory homemade

**Table 2 materials-16-05793-t002:** Static contact angles of representative surfaces at room temperature and after condensation.

Sample	Water Contact Angle (25 °C, RH = 30%)	Water Contact Angle (0 °C, RH = 70%)
Hydrophilic surface	22.1°	13.9°
Hydrophobic surface	137.3°	82.0°
Superhydrophobic surface 1 (Modified by anatase type nano titanium dioxide)	151.9°	147.8°
Superhydrophobic surface 2 (Modified by rutile type nano titanium dioxide)	154.0°	151.9°

## Data Availability

The data can be provided if needed by the corresponding author.

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
