# Peer review of "Anti-Condensation Performance of a New Superhydrophobic Coating for Pavements"

_materials, 2023, doi:10.3390/ma16175793_

Round 1

Reviewer 1 Report

It is an interesting article and its topic is of interest to pavement engineers. But the main problem in this article is the lack of presentation of results with figures, graphs and tables. The main question is whether this research is original and completely done by the authors? If the answer is positive, the article should be sent in full form and be reviewed again.

It needs minor corrections, especially that the sentences should be expressed anonymously (passive form). The word "we" is used a lot in the sentences.

Author Response

Answer: Thank you for your suggestions. We apologize that due to our negligence the first version was submitted without any figures, we have complemented missing information in the revised paper.

The manuscript has been uploaded, please see the attachment.

Reviewer 2 Report

The authors studied the effect of superhydrophobic coating in suppressing ice formation on pavements. The ice suppression ability is attributed to the ability of the coating in ensuring that the water droplet remains in the Cassie state. This is achieved controlling the surface roughness of the coating. 

I would like to note that the figures and tables are not included in the present manuscript, preventing a proper evaluation of the scientific value of the submission. The authors need to ensure that such trivial mistake is to be avoided for any manuscript submission.

In the present form, only a limited comments can be formed as follow:

Comment #1:

Please state the condition of the water used in the present experiments. Is it tap water? Deionized water? Also, how will the quality of the water affect the ice suppression ability of the coating?

Comment #2:

How can the authors prove that the water droplets remain in Cassie state in their experiments? Please elaborate and justify because this is explained to the underlying mechanism of the ice suppression ability of the coating. 

Comment #3:

In real life application, how can the nano-gap be maintained? Will external factors cause the gaps to change, resulting in the loss of ice suppression ability of the coating?

The language is acceptable and readable. It is to note that there are minor grammatical errors that need to be revised.

Author Response

The authors appreciate the reviewer for providing these valuable comments! The authors have carefully revised the manuscript based on those comments and suggestions. The detailed responses are listed below for your consideration:

Comment #1:

Please state the condition of the water used in the present experiments. Is it tap water? Deionized water? Also, how will the quality of the water affect the ice suppression ability of the coating?

Answer: Thanks for the suggestion. Based on the result of the reference ( Esmeryan, K.D. From Extremely Water-Repellent Coatings to Passive Icing Protection—Principles, Limitations and Innovative Application Aspects. Coatings, 2020), all kinds of liquids have large contact angles on the superhydrophobic surface, so the ice inhibition effect of the superhydrophobic coating also depends on different liquids. The superhydrophobic coating studied in this paper is mainly applied to the ice suppression of pavement. The icing of pavement is mainly affected by snow, ice and rain, which does not involve other liquids such as deionized water, oil or alcohol. So this paper does not carry out detailed research on this aspect. The water used in this experiment is tap water, which quality is similar to the snow, ice and rain.

Comment #2:

How can the authors prove that the water droplets remain in Cassie state in their experiments? Please elaborate and justify because this is explained to the underlying mechanism of the ice suppression ability of the coating. 

Answer: Thanks for the suggestion. From the water contact angle test, it is known that the contact angle of water droplets on different superhydrophobic surfaces is above 150° at a substrate temperature of 0°C. The contact angle does not change substantially with temperature and humidity. And the water droplets remain in the Cassie state in the presence of condensed water droplets.We can know that for a superhydrophobic surface, liquid water is located above the surface texture at room temperature, the air in the surface structure below the liquid water is in thermodynamic equilibrium with the liquid water. And then, the liquid water is in a Cassie state. In the ice adhesion strength test, liquid water on the surface gradually started to freeze when the temperature decreased. Subsequently the water droplets in the air did not freeze in the nanogap as the temperature decreased, but condensed and nucleated on the surface of the large water droplets. This indirectly proves that the water droplets at this time are also in the Cassie state. The actual contact area between the ice surface of the superhydrophobic surface and the specimen is small, which leads to the low adhesion strength of the ice-covered superhydrophobic surface in the experimental results. This issue is analyzed in subsection 4.4 of the article.

Comment #3:

In real life application, how can the nano-gap be maintained? Will external factors cause the gaps to change, resulting in the loss of ice suppression ability of the coating?

Answer: Thanks for the suggestion. The coatings prepared in this paper using existing proven processes can be experimentally verified that their nanogaps satisfy the conditions for condensation resistance. The external environment, such as dust, rain, and tire friction during driving, will have an effect on the surface nanostructure. Therefore, we will study the influence of external factors on the coating's ice inhibition performance through wear-resistant experiments and aging-resistant experiments. The research results will be published in subsequent papers.

The last manuscript has been uploaded, please the attachment.

Reviewer 3 Report

The topic seems interesting and presently images are not seen. It is very difficult to analyse the manuscript without images. I also see missing of few control experiments which can be beneficial towards the improvement of the manuscript also few references to self-cleaning technologies with diverse materials.

Adv. Funct. Mater. 2020, 1907772, ChemNanoMat 2023, e202300135, Nat. Rev. Mater. 2017, 2, 17036

Once the manuscript is updated with the images, references and enhanced discussion it can be considered Materials.

Needs to improve the language at several places of the manuscript 

Author Response

Answer: Sure, thanks for the suggestion. We apologize for not uploading the full article due to our negligence. The article has added figures, graphs, tables, references and enhanced discussion to present the results. At the same time, the whole manuscript has been double-checked in terms of language issues.  The  manuscript has been uploaded, please see the attachment.

Round 2

Reviewer 2 Report

The authors have answered the comments in an acceptable manner.

Language is acceptable. 

Author Response

The authors appreciate the reviewer for providing these valuable comments!

The new manuscript has been uploaded, thanks!

Reviewer 3 Report

The revised version with images looks ok. But major changes wrt figure captions, proper labelling of figures with sample names needs to be included. 

Is Figure 5 the SEM images of two different samples, as I see only one sample. Authors should clearly mention that. 

Write down the material name instead of superhydrophobic material surface

Figure 5. Field emission scanning electron microscope image of superhydrophobic material surface

This sentence is vague.

The micronano papillae were similar to the surface of lotus leaves.

Figure 5, SEM discussion is mostly baseless, needs through revamp wrt what authors notice from the images. 

Authors should clearly mention the figure captions unlike Figure 6. Before and after the water droplets freeze on the sample surface.

Is Figure 9. Adapted from some ref or why this image is wrt authors work?

I don’t see important references wrt self-cleaning technologies with diverse materials are included.

Adv. Funct. Mater. 2020, 30, 1907772, ChemNanoMat 2023, 9, e202300135, Philos. Trans. R. Soc., A 2019, 377, 20180270 

The interaction between TiO2 and SA needs to be highlighted.

I recommend major changeover. Without these corrections the manuscript is not suitable for the journal.

Authors should throughly check for the language corrections

Author Response

Comment #1:

Is Figure 5 the SEM images of two different samples, as I see only one sample. Authors should clearly mention that.

Write down the material name instead of superhydrophobic material surface

Figure 5. Field emission scanning electron microscope image of superhydrophobic material surface

Answer: Thanks for the suggestion. Figure 5 shows the field emission scanning electron microscopy images of the superhydrophobic surface modified with rutile titanium dioxide nanoparticles. (We discuss this part in section 4.1 of the article) Figure 5 (a) and (b) show the electron microscopy images at different magnifications.

Comment #2:

This sentence is vague.

The micronano papillae were similar to the surface of lotus leaves.

Figure 5, SEM discussion is mostly baseless, needs through revamp wrt what authors notice from the images.

Answer: Thanks for the suggestion. Lotus leaf is a natural superhydrophobic surface with self-cleaning characteristics. In this study, according to the characteristics of the superhydrophobic micro-nanostructures on the surface of lotus leaves[1], titanium dioxide, silicon dioxide and stearic acid were used as raw materials, compounded into a mixed dispersion and then centrifugally dried and ground into powder, and the superhydrophobic surfaces were prepared by spraying technology. In the electron microscope image, it can be observed that the superhydrophobic surface has the same papillae structure as the lotus leaf surface, thus confirming that the prepared samples are superhydrophobic surfaces.This has been explained in this article. (This section can be found on page 8.)

  • Bhushan B, Jung Y C. Natural and biomimetic artificial surfaces for superhydrophobicity, self-cleaning, low adhesion, and drag reduction. [J]. Progress in Materials Science , 2011, 56(1): 1-108.

Comment #3:

Authors should clearly mention the figure captions unlike Figure 6. Before and after the water droplets freeze on the sample surface.

Answer: Thanks for the suggestion. Image captions have been changed in the article.

Comment #4:

Is Figure 9. Adapted from some ref or why this image is wrt authors work?

Answer: Thanks for the suggestion.The figure is adapted from the following references.The image has been changed to the adapted form.

  • Yu Y B. Preparation and properties of superhydrophobic PVDF composite microporous membranes [D].Hebei University of Technology,2017.

Comment #5:

I don’t see important references wrt self-cleaning technologies with diverse materials are included.

Adv. Funct. Mater. 2020, 30, 1907772, ChemNanoMat 2023, 9, e202300135, Philos. Trans. R. Soc., A 2019, 377, 20180270 

Answer: Thanks for the suggestion.This part has already been added to the article.

Comment #6:

The interaction between TiO2 and SA needs to be highlighted.

Answer: Thanks for the suggestion. The heating reaction of stearic acid with anhydrous ethanol produces ethyl stearate with good lubricity and thermal stability, and the low surface energy substance produced by the reaction of stearic acid with ethanol constructs the rough structure of superhydrophobic materials. The hydrophobic modification effect of stearic acid on titanium dioxide nanoparticles played a decisive role in the hydrophobicity of the samples. This interpretation is reinforced in the article. (This section can be found on page 5.)

The last manuscript has been uploaded, thanks!
